# STABILIZING ADVERSARIAL NETS WITH PREDICTION METHODS

**Abhay Yadav,**\* **Sohil Shah,**\* **Zheng Xu, David Jacobs, & Tom Goldstein**
University of Maryland,
College Park, MD 20740, USA.
`{jaiabhay, xuzh, tomg}@cs.umd.edu, sohilas@umd.edu,`
`djacobs@umiacs.umd.edu`

## ABSTRACT

Adversarial neural networks solve many important problems in data science, but are notoriously difficult to train. These difficulties come from the fact that optimal weights for adversarial nets correspond to saddle points, and not minimizers, of the loss function. The alternating stochastic gradient methods typically used for such problems do not reliably converge to saddle points, and when convergence does happen it is often highly sensitive to learning rates. We propose a simple modification of stochastic gradient descent that stabilizes adversarial networks. We show, both in theory and practice, that the proposed method reliably converges to saddle points, and is stable with a wider range of training parameters than a non-prediction method. This makes adversarial networks less likely to "collapse," and enables faster training with larger learning rates.

## 1 INTRODUCTION

Adversarial networks play an important role in a variety of applications, including image generation (Zhang et al., 2017; Wang & Gupta, 2016), style transfer (Brock et al., 2017; Taigman et al., 2017; Wang & Gupta, 2016; Isola et al., 2017), domain adaptation (Taigman et al., 2017; Tzeng et al., 2017; Ganin & Lempitsky, 2015), imitation learning (Ho et al., 2016), privacy (Edwards & Storkey, 2016; Abadi & Andersen, 2016), fair representation (Mathieu et al., 2016; Edwards & Storkey, 2016), etc. One particularly motivating application of adversarial nets is their ability to form generative models, as opposed to the classical discriminative models (Goodfellow et al., 2014; Radford et al., 2016; Denton et al., 2015; Mirza & Osindero, 2014).

While adversarial networks have the power to attack a wide range of previously unsolved problems, they suffer from a major flaw: they are difficult to train. This is because adversarial nets try to accomplish two objectives simultaneously; weights are adjusted to maximize performance on one task while minimizing performance on another. Mathematically, this corresponds to finding a *saddle point* of a loss function - a point that is minimal with respect to one set of weights, and maximal with respect to another.

Conventional neural networks are trained by marching down a loss function until a minimizer is reached (Figure 1a). In contrast, adversarial training methods search for saddle points rather than a minimizer, which introduces the possibility that the training path "slides off" the objective functions and the loss goes to $-\infty$ (Figure 1b), resulting in "collapse" of the adversarial network. As a result, many authors suggest using early stopping, gradients/weight clipping (Arjovsky et al., 2017), or specialized objective functions (Goodfellow et al., 2014; Zhao et al., 2017; Arjovsky et al., 2017) to maintain stability.

In this paper, we present a simple "prediction" step that is easily added to many training algorithms for adversarial nets. We present theoretical analysis showing that the proposed prediction method is asymptotically stable for a class of saddle point problems. Finally, we use a wide range of experiments to show that prediction enables faster training of adversarial networks using large learning rates without the instability problems that plague conventional training schemes.

---

\*Equal contribution

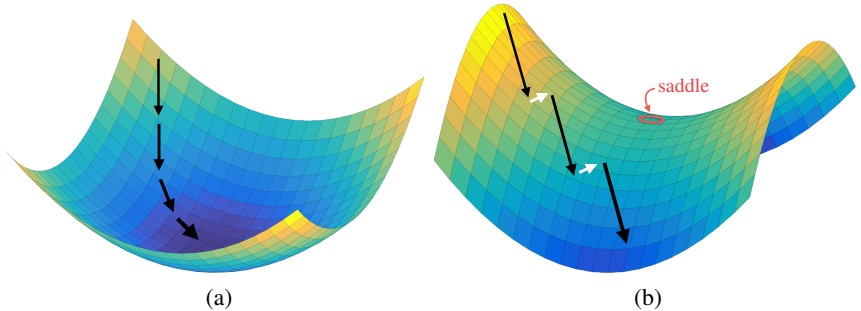

(a)           (b)

Figure 1: A schematic depiction of gradient methods. (a) Classical networks are trained by marching down the loss function until a minimizer is reached. Because classical loss functions are bounded from below, the solution path gets stopped when a minimizer is reached, and the gradient method remains stable. (b) Adversarial net loss functions may be unbounded from below, and training alternates between minimization and maximization steps. If minimization (or, conversely, maximization) is more powerful, the solution path "slides off" the loss surface and the algorithm becomes unstable, resulting in a sudden "collapse" of the network.

## 2   PROPOSED METHOD

Saddle-point optimization problems have the general form

$$\min_u \max_v \mathcal{L}(u, v) \tag{1}$$

for some loss function $\mathcal{L}$ and variables $u$ and $v$. Most authors use the alternating stochastic gradient method to solve saddle-point problems involving neural networks. This method alternates between updating $u$ with a stochastic gradient *descent* step, and then updating $v$ with a stochastic gradient *ascent* step. When simple/classical SGD updates are used, the steps of this method can be written

$$
\begin{aligned}
u^{k+1} &= u^k - \alpha_k \mathcal{L}'_u(u^k, v^k) \quad | \quad \text{gradient descent in } u, \text{ starting at } (u^k, v^k) \\
v^{k+1} &= v^k + \beta_k \mathcal{L}'_v(u^{k+1}, v^k) \quad | \quad \text{gradient ascent in } v, \text{ starting at } (u^{k+1}, v^k).
\end{aligned}
\tag{2}
$$

Here, $\{\alpha_k\}$ and $\{\beta_k\}$ are learning rate schedules for the minimization and maximization steps, respectively. The vectors $\mathcal{L}'_u(u, v)$ and $\mathcal{L}'_v(u, v)$ denote (possibly stochastic) gradients of $\mathcal{L}$ with respect to $u$ and $v$. In practice, the gradient updates are often performed by an automated solver, such as the Adam optimizer (Kingma & Ba, 2015), and include momentum updates.

We propose to stabilize the training of adversarial networks by adding a *prediction* step. Rather than calculating $v^{k+1}$ using $u^{k+1}$, we first make a prediction, $\bar{u}^{k+1}$, about where the $u$ iterates will be in the future, and use this predicted value to obtain $v^{k+1}$.

---

**Prediction Method**

$$
\begin{aligned}
u^{k+1} &= u^k - \alpha_k \mathcal{L}'_u(u^k, v^k) \quad | \quad \text{gradient descent in } u, \text{ starting at } (u^k, v^k) \\
\bar{u}^{k+1} &= u^{k+1} + (u^{k+1} - u^k) \quad | \quad \textit{predict} \text{ future value of } u \\
v^{k+1} &= v^k + \beta_k \mathcal{L}'_v(\bar{u}^{k+1}, v^k) \quad | \quad \text{gradient ascent in } v, \text{ starting at } (\bar{u}^{k+1}, v^k).
\end{aligned}
\tag{3}
$$

---

The Prediction step (3) tries to estimate where $u$ is going to be in the future by assuming its trajectory remains the same as in the current iteration.

## 3   BACKGROUND

### 3.1   ADVERSARIAL NETWORKS AS A SADDLE-POINT PROBLEM

We now discuss a few common adversarial network problems and their saddle-point formulations. *Generative Adversarial Networks* (GANs) fit a generative model to a dataset using a game in which a generative model competes against a discriminator (Goodfellow et al., 2014). The generator,

$\mathbf{G}(\mathbf{z}; \theta_g)$, takes random noise vectors $\mathbf{z}$ as inputs, and maps them onto points in the target data distribution. The discriminator, $\mathbf{D}(\mathbf{x}; \theta_d)$, accepts a candidate point $\mathbf{x}$ and tries to determine whether it is really drawn from the empirical distribution (in which case it outputs 1), or fabricated by the generator (output 0). During a training iteration, noise vectors from a Gaussian distribution $\mathcal{G}$ are pushed through the generator network $\mathbf{G}$ to form a batch of generated data samples denoted by $\mathcal{D}_{fake}$. A batch of empirical samples, $\mathcal{D}_{real}$, is also prepared. One then tries to adjust the weights of each network to solve a saddle point problem, which is popularly formulated as,

$$\min_{\theta_g} \max_{\theta_d} \quad \mathbb{E}_{x \sim \mathcal{D}_{real}} f(\mathbf{D}(\mathbf{x}; \theta_d)) + \mathbb{E}_{z \sim \mathcal{G}} f(1 - \mathbf{D}(\mathbf{G}(\mathbf{z}; \theta_g); \theta_d)). \quad (4)$$

Here $f(.)$ is any monotonically increasing function. Initially, (Goodfellow et al., 2014) proposed using $f(x) = \log(x)$.

*Domain Adversarial Networks* (DANs) (Makhzani et al., 2016; Ganin & Lempitsky, 2015; Edwards & Storkey, 2016) take data collected from a "source" domain, and extract a feature representation that can be used to train models that generalize to another "target" domain. For example, in the domain adversarial neural network (DANN (Ganin & Lempitsky, 2015)), a set of feature layers maps data points into an embedded feature space, and a classifier is trained on these embedded features. Meanwhile, the adversarial discriminator tries to determine, using only the embedded features, whether the data points belong to the source or target domain. A good embedding yields a better task-specific objective on the target domain while fooling the discriminator, and is found by solving

$$\min_{\theta_f, \theta_{y^k}} \max_{\theta_d} \quad \sum_k \alpha_k \mathcal{L}_{y^k} \left( \mathbf{x}_s; \theta_f, \theta_{y^k} \right) - \lambda \mathcal{L}_d \left( \mathbf{x}_s, \mathbf{x}_t; \theta_f, \theta_d \right). \quad (5)$$

Here $\mathcal{L}_d$ is any adversarial discriminator loss function and $\mathcal{L}_{y^k}$ denotes the task specific loss. $\theta_f, \theta_d,$ and $\theta_{y^k}$ are network parameter of feature mapping, discriminator, and classification layers.

## 3.2 STABILIZING SADDLE POINT SOLVERS

It is well known that alternating stochastic gradient methods are unstable when using simple logarithmic losses. This led researchers to explore multiple directions for stabilizing GANs; either by adding regularization terms (Arjovsky et al., 2017; Li et al., 2015; Che et al., 2017; Zhao et al., 2017), a myriad of training "hacks" (Salimans et al., 2016; Gulrajani et al., 2017), re-engineering network architectures (Zhao et al., 2017), and designing different solvers (Metz et al., 2017). Specifically, the Wasserstein GAN (WGAN) (Arjovsky et al., 2017) approach modifies the original objective by replacing $f(x) = \log(x)$ with $f(x) = x$. This led to a training scheme in which the discriminator weights are "clipped." However, as discussed in Arjovsky et al. (2017), the WGAN training is unstable at high learning rates, or when used with popular momentum based solvers such as Adam. Currently, it is known to work well only with RMSProp (Arjovsky et al., 2017).

The unrolled GAN (Metz et al., 2017) is a new solver that can stabilize training at the cost of more expensive gradient computations. Each generator update requires the computation of multiple extra discriminator updates, which are then discarded when the generator update is complete. While avoiding GAN collapse, this method requires increased computation and memory.

In the convex optimization literature, saddle point problems are more well studied. One popular solver is the primal-dual hybrid gradient (PDHG) method (Zhu & Chan, 2008; Esser et al., 2009), which has been popularized by Chambolle and Pock (Chambolle & Pock, 2011), and has been successfully applied to a range of machine learning and statistical estimation problems (Goldstein et al., 2015). PDHG relates closely to the method proposed here - it achieves stability using the same prediction step, although it uses a different type of gradient update and is only applicable to bi-linear problems.

Stochastic methods for convex saddle-point problems can be roughly divided into two categories: stochastic coordinate descent (Dang & Lan, 2014; Lan & Zhou, 2015; Zhang & Lin, 2015; Zhu & Storkey, 2015; 2016; Wang & Xiao, 2017; Shibagaki & Takeuchi, 2017) and stochastic gradient descent (Chen et al., 2014; Qiao et al., 2016). Similar optimization algorithms have been studied for reinforcement learning (Wang & Chen, 2016; Du et al., 2017). Recently, a "doubly" stochastic method that randomizes both primal and dual updates was proposed for strongly convex bilinear saddle point problems (Yu et al., 2015). For general saddle point problems, "doubly" stochastic gradient descent methods are discussed in Nemirovski et al. (2009),Palaniappan & Bach (2016), in

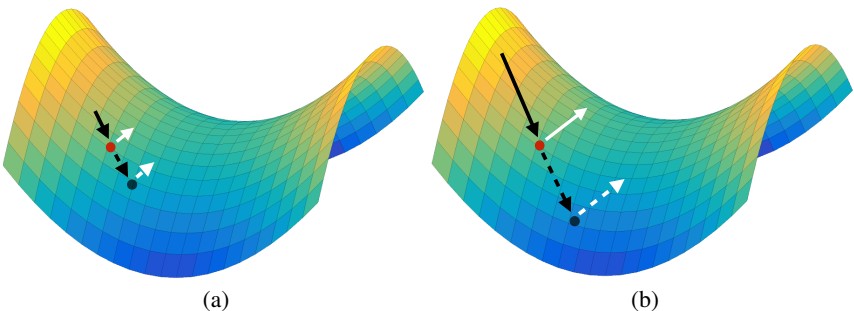

(a)                                                      (b)

Figure 2: A schematic depiction of the prediction method. When the minimization step is powerful and moves the iterates a long distance, the prediction step (dotted black arrow) causes the maximization update to be calculated further down the loss surface, resulting in a more dramatic maximization update. In this way, prediction methods prevent the maximization step from getting overpowered by the minimization update.

which primal and dual variables are updated simultaneously based on the previous iterates and the current gradients.

## 4    INTERPRETATIONS OF THE PREDICTION STEP

We present three ways to explain the effect of prediction: an intuitive, non-mathematical perspective, a more analytical viewpoint involving dynamical systems, and finally a rigorous proof-based approach.

### 4.1    AN INTUITIVE VIEWPOINT

The standard alternating SGD switches between minimization and maximization steps. In this algorithm, there is a risk that the minimization step can overpower the maximization step, in which case the iterates will "slide off" the edge of saddle, leading to instability (Figure 1b). Conversely, an overpowering maximization step will dominate the minimization step, and drive the iterates to extreme values as well.

The effect of prediction is visualized in Figure 2. Suppose that a maximization step takes place starting at the red dot. Without prediction, the maximization step has no knowledge of the algorithm history, and will be the same regardless of whether the previous minimization update was weak (Figure 2a) or strong (Figure 2b). Prediction allows the maximization step to exploit information about the minimization step. If the previous minimizations step was weak (Figure 2a), the prediction step (dotted black arrow) stays close to the red dot, resulting in a weak predictive maximization step (white arrow). But if we arrived at the red dot using a strong minimization step (Figure 2b), the prediction moves a long way down the loss surface, resulting in a stronger maximization step (white arrows) to compensate.

### 4.2    A MORE MATHEMATICAL PERSPECTIVE

To get stronger intuition about prediction methods, let's look at the behavior of Algorithm (3) on a simple bi-linear saddle of the form

$$\mathcal{L}(u, v) = v^T K u \tag{6}$$

where $K$ is a matrix. When exact (non-stochastic) gradient updates are used, the iterates follow the path of a simple dynamical system with closed-form solutions. We give here a sketch of this argument: a detailed derivation is provided in the Supplementary Material.

When the (non-predictive) gradient method (2) is applied to the linear problem (6), the resulting iterations can be written

$$\frac{u^{k+1} - u^k}{\alpha} = -K^T v^k, \qquad \frac{v^{k+1} - v^k}{\alpha} = (\beta/\alpha) K u^{k+1}.$$

When the stepsize $\alpha$ gets small, this behaves like a discretization of the system of differential equations

$$\dot{u} = -K^T v, \qquad \dot{v} = \beta/\alpha K u$$

where $\dot{u}$ and $\dot{v}$ denote the derivatives of $u$ and $v$ with respect to time. These equations describe a simple harmonic oscillator, and the closed form solution for $u$ is

$$u(t) = C\cos(\Sigma^{1/2}t + \phi)$$

where $\Sigma$ is a diagonal matrix, and the matrix $C$ and vector $\phi$ depend on the initialization. We can see that, for small values of $\alpha$ and $\beta$, the non-predictive algorithm (2) approximates an undamped harmonic motion, and the solutions orbit around the saddle without converging.

The prediction step (3) improves convergence because it produces *damped* harmonic motion that sinks into the saddle point. When applied to the linearized problem (6), we get the dynamical system

$$\dot{u} = -K^T v, \qquad \dot{v} = \beta/\alpha K(u + \alpha\dot{u}) \tag{7}$$

which has solution

$$u(t) = UA\exp(-\frac{t\alpha}{2}\sqrt{\Sigma})\sin(t\sqrt{(1 - \alpha^2/4)\Sigma} + \phi).$$

From this analysis, we see that the damping caused by the prediction step causes the orbits to converge into the saddle point, and the error decays exponentially fast.

### 4.3 A RIGOROUS PERSPECTIVE

While the arguments above are intuitive, they are also informal and do not address issues like stochastic gradients, non-constant stepsize sequences, and more complex loss functions. We now provide a rigorous convergence analysis that handles these issues.

We assume that the function $\mathcal{L}(u, v)$ is convex in $u$ and concave in $v$. We can then measure convergence using the "primal-dual" gap, $P(u, v) = \mathcal{L}(u, v^\star) - \mathcal{L}(u^\star, v)$ where $(u^\star, v^\star)$ is a saddle. Note that $P(u, v) > 0$ for non-optimal $(u, v)$, and $P(u, v) = 0$ if $(u, v)$ is a saddle. Using these definitions, we formulate the following convergence result. The proof is in the supplementary material.

**Theorem 1.** *Suppose the function $\mathcal{L}(u, v)$ is convex in $u$, concave in $v$, and that the partial gradient $\mathcal{L}_v'$ is uniformly Lipschitz smooth in $u$ ($\|\mathcal{L}_v'(u_1, v) - \mathcal{L}_v'(u_2, v)\| \le L_v\|u_1 - u_2\|$). Suppose further that the stochastic gradient approximations satisfy $\mathbb{E}\|\mathcal{L}_u'(u, v)\|^2 \le G_u^2$, $\mathbb{E}\|\mathcal{L}_v'(u, v)\|^2 \le G_v^2$ for scalars $G_u$ and $G_v$, and that $\mathbb{E}\|u^k - u^\star\|^2 \le D_u^2$, and $\mathbb{E}\|v^k - v^\star\|^2 \le D_v^2$ for scalars $D_u$ and $D_v$.*

*If we choose decreasing learning rate parameters of the form $\alpha_k = \frac{C_\alpha}{\sqrt{k}}$ and $\beta_k = \frac{C_\beta}{\sqrt{k}}$, then the SGD method with prediction converges in expectation, and we have the error bound*

$$\mathbb{E}[P(\hat{u}^l, \hat{v}^l)] \le \frac{1}{2\sqrt{l}}\left(\frac{D_u^2}{C_\alpha} + \frac{D_v^2}{C_\beta}\right) + \frac{\sqrt{l+1}}{l}\left(\frac{C_\alpha G_u^2}{2} + C_\alpha L_v G_u^2 + C_\alpha L_v D_v^2 + \frac{C_\beta G_v^2}{2}\right)$$

*where $\hat{u}^l = \frac{1}{l}\sum_{k=1}^{l} u^k$, $\hat{v}^l = \frac{1}{l}\sum_{k=1}^{l} v^k$.*

## 5 EXPERIMENTS

We present a wide range of experiments to demonstrate the benefits of the proposed prediction step for adversarial nets. We consider a saddle point problem on a toy dataset constructed using MNIST images, and then move on to consider state-of-the-art models for three tasks: GANs, domain adaptation, and learning of fair classifiers. Additional results, and additional experiments involving mixtures of Gaussians, are presented in the Appendix. The code is available at `https://github.com/jaiabhayk/stableGAN`.

### 5.1 MNIST TOY PROBLEM

We consider the task of classifying MNIST digits as being even or odd. To make the problem interesting, we corrupt 70% of odd digits with salt-and-pepper noise, while we corrupt only 30% of even digits. When we train a LeNet network (LeCun et al., 1998) on this problem, we find that the network encodes and uses information about the noise; when a noise vs no-noise classifier is trained

on the deep features generated by LeNet, it gets 100% accuracy. The goal of this task is to force LeNet to ignore the noise when making decisions. We create an adversarial model of the form (5) in which $\mathcal{L}_y$ is a softmax loss for the even vs odd classifier. We make $\mathcal{L}_d$ a softmax loss for the task of discriminating whether the input sample is noisy or not. The classifier and discriminator were both pre-trained using the default LeNet implementation in Caffe (Jia et al., 2014). Then the combined adversarial net was jointly trained both with and without prediction. For implementation details, see the Supplementary Material.

Figure 3 summarizes our findings. In this experiment, we considered applying prediction to both the classifier and discriminator. We note that our task is to retain good classification accuracy while preventing the discriminator from doing better than the trivial strategy of classifying odd digits as noisy and even digits as non-noisy. This means that the discriminator accuracy should ideally be $\sim 0.7$. As shown in Figure 3a, the prediction step hardly makes any difference when evaluated at the small learning rate of $10^{-4}$. However, when evaluated at higher rates, Figures 3b and 3c show that the prediction solvers are very stable while one without prediction collapses (blue solid line is flat) very early. Figure 3c shows that the default learning rate ($10^{-3}$) of the Adam solver is unstable unless prediction is used.

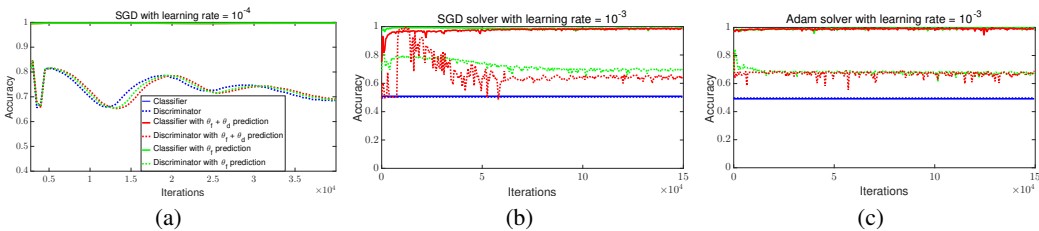

Figure 3: Comparison of the classification accuracy (digit parity) and discriminator (noisy vs. no-noise) accuracy using SGD and Adam solver with and without prediction steps. $\theta_f$ and $\theta_d$ refers to variables in eq. (5). (a) Using SGD with learning rate $lr = 10^{-4}$. Note that the solid lines of red, blue and green are overlapped. (b) SGD solver with higher learning rate of $lr = 10^{-3}$, and (c) using Adam solver with its default parameter.

## 5.2 GENERATIVE ADVERSARIAL NETWORKS

Next, we test the efficacy and stability of our proposed predictive step on generative adversarial networks (GAN), which are formulated as saddle point problems (4) and are popularly solved using a heuristic approach (Goodfellow et al., 2014). We consider an image modeling task using CIFAR-10 (Krizhevsky, 2009) on the recently popular convolutional GAN architecture, DCGAN (Radford et al., 2016). We compare our predictive method with that of DCGAN and the unrolled GAN (Metz et al., 2017) using the training protocol described in Radford et al. (2016). Note that we compared against the unrolled GAN with stop gradient switch[1] and $K = 5$ unrolling steps. All the approaches were trained for five random seeds and 100 epochs each.

We start with comparing all three methods using the default solver for DCGAN (the Adam optimizer) with learning rate=0.0002 and $\beta_1$=0.5. Figure 4 compares the generated sample images (at the $100^{th}$ epoch) and the training loss curve for all approaches. The discriminator and generator loss curves in Figure 4e show that without prediction, the DCGAN collapses at the $45^{th}$ and $57^{th}$ epochs. Similarly, Figure 4f shows that the training for unrolled GAN collapses in at least three instances. The training procedure using predictive steps never collapsed during any epochs. Qualitatively, the images generated using prediction are more diverse than the DCGAN and unrolled GAN images.

Figure 5 compares all approaches when trained with $5\times$ higher learning rate (0.001) (the default for the Adam solver). As observed in Radford et al. (2016), the standard and unrolled solvers are very unstable and collapse at this higher rate. However, as shown in Figure 5d, & 5a, training remains stable when a predictive step is used, and generates images of reasonable quality. The training procedure for both DCGAN and unrolled GAN collapsed on all five random seeds. The results on various additional intermediate learning rates as well as on high resolution Imagenet dataset are in the Supplementary Material.

---

[1] We found the unrolled GAN without stop gradient switch as well as for smaller values of $K$ collapsed when used on the DCGAN architecture.

In the Supplementary Material, we present one additional comparison showing results on a higher momentum of $\beta_1$=0.9 (learning rate=0.0002). We observe that all the training approaches are stable. However, the quality of images generated using DCGAN is inferior to that of the predictive and unrolled methods.

Overall, of the 25 training settings we ran on (each of five learning rates for five random seeds), the DCGAN training procedure collapsed in 20 such instances while unrolled GAN collapsed in 14 experiments (not counting the multiple collapse in each training setting). On the contrary, we find that our simple predictive step method collapsed only once.

Note that prediction adds trivial cost to the training algorithm. Using a single TitanX Pascal, a training epoch of DCGAN takes 35 secs. With prediction, an epoch takes 38 secs. The unrolled GAN method, which requires extra gradient steps, takes 139 secs/epoch.

Finally, we draw quantitative comparisons based on the inception score (Salimans et al., 2016), which is a widely used metric for visual quality of the generated images. For this purpose, we consider the current state-of-the-art Stacked GAN (Huang et al., 2017) architecture. Table 1 lists the inception scores computed on the generated samples from Stacked GAN trained (200 epochs) with and without prediction at different learning rates. The joint training of Stacked GAN collapses when trained at the default learning rate of adam solver (i.e., 0.001). However, reasonably good samples are generated if the same is trained with prediction on both the generator networks. The right end of Table 1 also list the inception score measured at fewer number of epochs for higher learning rates. It suggest that the model trained with prediction methods are not only stable but also allows faster convergence using higher learning rates. For reference the inception score on real images of CIFAR-10 dataset is $11.51 \pm 0.17$.

Table 1: Comparison of Inception Score on Stacked GAN network with and w/o **G** prediction.

| Learning rate | 0.0001 | 0.0005 | 0.001 | 0.0005 (40) | 0.001 (20) |
|---|---|---|---|---|---|
| Stacked GAN (joint) | $8.44 \pm 0.11$ | $7.90 \pm 0.08$ | $1.52 \pm 0.01$ | $5.80 \pm 0.15$ | $1.42 \pm 0.01$ |
| Stacked GAN (joint) + prediction | $\mathbf{8.55 \pm 0.12}$ | $\mathbf{8.13 \pm 0.09}$ | $\mathbf{7.96 \pm 0.11}$ | $\mathbf{8.10 \pm 0.10}$ | $\mathbf{7.79 \pm 0.07}$ |

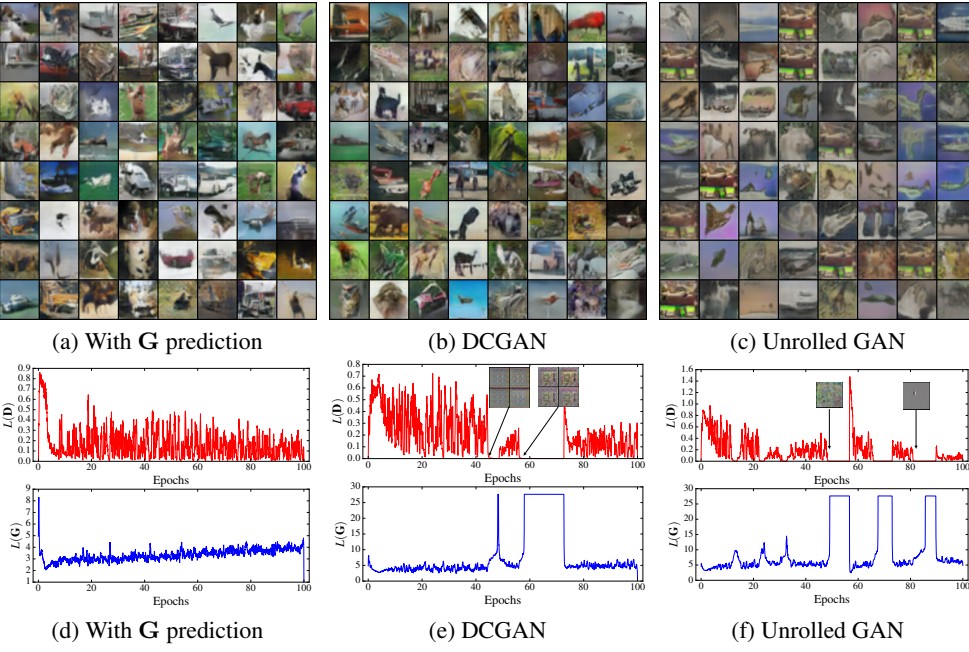

(a) With **G** prediction     (b) DCGAN     (c) Unrolled GAN

(d) With **G** prediction     (e) DCGAN     (f) Unrolled GAN

Figure 4: Comparison of GAN training algorithms for DCGAN architecture on Cifar-10 image datasets. Using default parameters of DCGAN; $lr = 0.0002, \beta_1 = 0.5$.

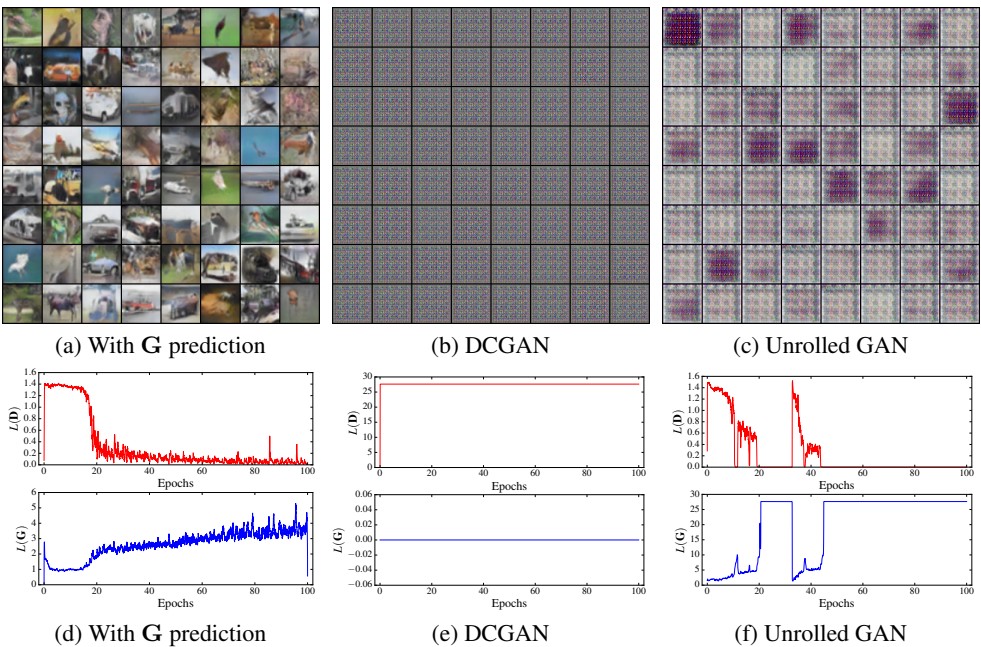

| (a) With **G** prediction | (b) DCGAN | (c) Unrolled GAN |
|---|---|---|
| (d) With **G** prediction | (e) DCGAN | (f) Unrolled GAN |

Figure 5: Comparison of GAN training algorithms for DCGAN architecture on Cifar-10 image datasets with higher learning rate, $lr = 0.001, \beta_1 = 0.5$.

## 5.3 DOMAIN ADAPTATION

We consider the domain adaptation task (Saenko et al., 2010; Ganin & Lempitsky, 2015; Tzeng et al., 2017) wherein the representation learned using the source domain samples is altered so that it can also generalize to samples from the target distribution. We use the problem setup and hyper-parameters as described in (Ganin & Lempitsky, 2015) using the OFFICE dataset (Saenko et al., 2010) (experimental details are shared in the Supplementary Material). In Table 2, comparisons are drawn with respect to target domain accuracy on six pairs of source-target domain tasks. We observe that the prediction step has mild benefits on the "easy" adaptation tasks with very similar source and target domain samples. However, on the transfer learning tasks of AMAZON-to-WEBCAM, WEBCAM-to-AMAZON, and DSLR-to-AMAZON which has noticeably distinct data samples, an extra prediction step gives an absolute improvement of $1.3 - 6.9\%$ in predicting target domain labels.

Table 2: Comparison of target domain accuracy on OFFICE dataset.

| Method | Source Target | AMAZON WEBCAM | WEBCAM AMAZON | DSLR WEBCAM | WEBCAM DSLR | AMAZON DSLR | DSLR AMAZON |
|---|---|---|---|---|---|---|---|
| DANN (Ganin & Lempitsky, 2015) | | 73.4 | 51.6 | 95.5 | **99.4** | **76.5** | 51.7 |
| DANN + prediction | | **74.7** | **58.5** | **96.1** | 99.0 | 73.5 | **57.6** |

## 5.4 FAIR CLASSIFIER

Finally, we consider a task of learning fair feature representations (Mathieu et al., 2016; Edwards & Storkey, 2016; Louizos et al., 2016) such that the final learned classifier does not discriminate with respect to a sensitive variable. As proposed in Edwards & Storkey (2016) one way to measure fairness is using discrimination,

$$y_{disc} = \left| \frac{1}{N_0} \sum_{i:s_i=0} \eta(x_i) - \frac{1}{N_1} \sum_{i:s_i=1} \eta(x_i) \right|. \tag{8}$$

Here $s_i$ is a binary sensitive variable for the $i^{th}$ data sample and $N_k$ denotes the total number of samples belonging to the $k^{th}$ sensitive class. Similar to the domain adaptation task, the learning of each classifier can be formulated as a minimax problem in (5) (Edwards & Storkey, 2016; Mathieu

et al., 2016). Unlike the previous example though, this task has a model selection component. From a pool of hundreds of randomly generated adversarial deep nets, for each value of $t$, one selects the model that maximizes the difference

$$y_{t,Delta} = y_{acc} - t * y_{disc}. \tag{9}$$

The "Adult" dataset from the UCI machine learning repository is used. The task ($y_{acc}$) is to classify whether a person earns $\geq \$50k$/year. The person's gender is chosen to be the sensitive variable. Details are in the supplementary. To demonstrate the advantage of using prediction for model selection, we follow the protocol developed in Edwards & Storkey (2016). In this work, the search space is restricted to a class of models that consist of a fully connected autoencoder, one task specific discriminator, and one adversarial discriminator. The encoder output from the autoencoder acts as input to both the discriminators. In our experiment, 100 models are randomly selected. During the training of each adversarial model, $\mathcal{L}_d$ is a cross-entropy loss while $\mathcal{L}_y$ is a linear combination of reconstruction and cross-entropy loss. Once all the models are trained, the best model for each value of $t$ is selected by evaluating (9) on the validation set.

Figure 6a plots the results on the test set for the AFLR approach with and without prediction steps in their default Adam solver. For each value of $t$, Figure 6b, 6c also compares the number of layers in the selected encoder and discriminator networks. When using prediction for training, relatively stronger encoder models are produced and selected during validation, and hence the prediction results generalize better on the test set.

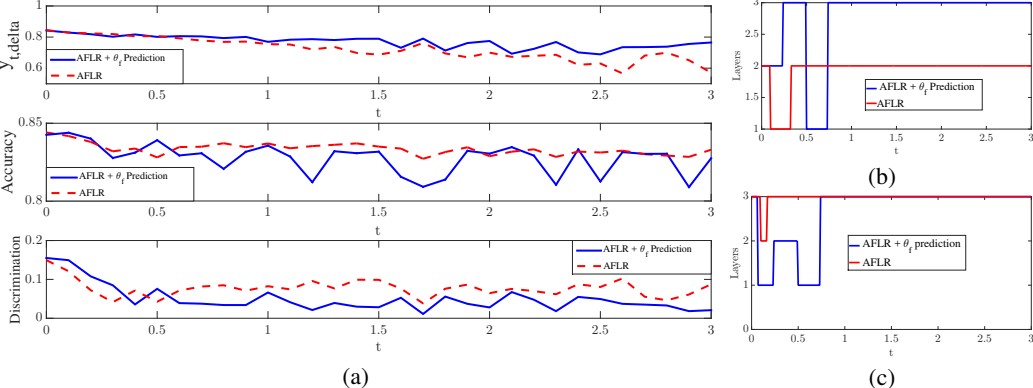

(a)                                                    (b)

                                                       (c)

Figure 6: Model selection for learning a fair classifier. (a) Comparison of $y_{t,delta}$ (higher is better), and also $y_{disc}$ (lower is better) and $y_{acc}$ on the test set using AFLR with and without predictive steps. (b) Number of encoder layers in the selected model. (c) Number of discriminator layers (both adversarial and task-specific) in the selected model.

## 6    CONCLUSION

We present a simple modification to the alternating SGD method, called a prediction step, that improves the stability of adversarial networks. We present theoretical results showing that the prediction step is asymptotically stable for solving saddle point problems. We show, using a variety of test problems, that prediction steps prevent network collapse and enable training with a wider range of learning rates than plain SGD methods.

### ACKNOWLEDGMENTS

The work of T. Goldstein was supported by the US Office of Naval Research under grant N00014-17-1-2078, the US National Science Foundation (NSF) under grant CCF-1535902, and by the Sloan Foundation. A. Yadav and D. Jacobs were supported by the National Science Foundation under grant no. IIS-1526234 and by the Office of the Director of National Intelligence (ODNI), Intelligence Advanced Research Projects Activity (IARPA), via IARPA R&D Contract No. 2014-14071600012. The views and conclusions contained herein are those of the authors and should not be interpreted as necessarily representing the official policies or endorsements, either expressed or implied, of

the ODNI, IARPA, or the U.S. Government. The U.S. Government is authorized to reproduce and distribute reprints for Governmental purposes notwithstanding any copyright annotation thereon.

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

APPENDIX

## A    DETAILED DERIVATION OF THE HARMONIC OSCILLATOR EQUATION

Here, we provide a detailed derivation of the harmonic oscillator behavior of Algorithm (3) on the simple bi-linear saddle of the form

$$\mathcal{L}(x, y) = y^T K x$$

where $K$ is a matrix. Note that, within a small neighborhood of a saddle, all smooth weakly convex objective functions behave like (6).To see why, consider a smooth objective function $\mathcal{L}$ with a saddle point at $x^* = 0$, $y^* = 0$. Within a small neighborhood of the saddle, we can approximate the function $\mathcal{L}$ to high accuracy using its Taylor approximation

$$\mathcal{L}(x, y) \approx \mathcal{L}(x^*, y^*) + y^T \mathcal{L}'_{xy} x + O(\|x\|^3 + \|y\|^3)$$

where $\mathcal{L}'_{xy}$ denotes the matrix of mixed-partial derivatives with respect to $x$ and $y$. Note that the first-order terms have vanished from this Taylor approximation because the gradients are zero at a saddle point. The $O(\|x\|^2)$ and $O(\|y\|^2)$ terms vanish as well because the problem is assumed to be weakly convex around the saddle. Up to third-order error (which vanishes quickly near the saddle), this Taylor expansion has the form (6). For this reason, stability on saddles of the form (6) is a necessary condition for convergence of (3), and the analysis here describes the asymptotic behavior of the prediction method on any smooth problem for which the method converges.

We will show that, as the learning rate gets small, the iterates of the non-prediction method (2) rotate in orbits around the saddle without converging. In contrast, the iterates of the prediction method fall into the saddle and converge.

When the conventional gradient method (2) is applied to the linear problem (6), the resulting iterations can be written

$$\frac{x^{k+1} - x^k}{\alpha} = -K^T y^k, \qquad \frac{y^{k+1} - y^k}{\alpha} = (\beta/\alpha) K x^{k+1}.$$

When the stepsize $\alpha$ gets small, this behaves like a discretization of the differential equation

$$\dot{x} = -K^T y \tag{10}$$
$$\dot{y} = \beta/\alpha K x \tag{11}$$

where $\dot{x}$ and $\dot{y}$ denote the derivatives of $x$ and $y$ with respect to time.

The differential equations (10,11) describe a harmonic oscillator. To see why, differentiate (10) and plug (11) into the result to get a differential equation in $x$ alone

$$\ddot{x} = -K^T \dot{y} = -\beta/\alpha K^T K x. \tag{12}$$

We can decompose this into a system of independent single-variable problems by considering the eigenvalue decomposition $\beta/\alpha K^T K = U\Sigma U^T$. We now multiply both sides of (12) by $U^T$, and make the change of variables $z \leftarrow U^T x$ to get

$$\ddot{z} = -\Sigma z.$$

where $\Sigma$ is diagonal. This is the standard equation for undamped harmonic motion, and its solution is $z = A \cos(\Sigma^{1/2} t + \phi)$, where $\cos$ acts entry-wise, and the diagonal matrix $A$ and vector $\phi$ are constants that depend only on the initialization. Changing back into the variable $x$, we get the solution

$$x = UA \cos(\Sigma^{1/2} t + \phi).$$

We can see that, for small values of $\alpha$ and $\beta$, the non-predictive algorithm (2) approximates an undamped harmonic motion, and the solutions orbit around the saddle without converging.

The prediction step (3) improves convergence because it produces *damped* harmonic motion that sinks into the saddle point. When applied to the linearized problem (6), the iterates of the predictive method (3) satisfy

$$\frac{x^{k+1} - x^k}{\alpha} = -K^T y^k$$
$$\frac{y^{k+1} - y^k}{\alpha} = \beta/\alpha K(x^{k+1} + x^{k+1} - x^k) = \beta/\alpha K x^{k+1} + \beta K \frac{x^{k+1} - x^k}{\alpha}.$$

For small $\alpha$, this approximates the dynamical system

$$\dot{x} = -K^T y \tag{13}$$
$$\dot{y} = \beta/\alpha K(x + \alpha \dot{x}). \tag{14}$$

Like before, we differentiate (13) and use (14) to obtain

$$\ddot{x} = -K^T \dot{y} = -\beta/\alpha K^T (Kx + \alpha A\dot{x}) = -\beta/\alpha K^T Kx - \beta/K^T K\dot{x}. \tag{15}$$

Finally, multiply both sides by $U^T$ and perform the change of variables $z \leftarrow U^T x$ to get

$$\ddot{z} = -\Sigma z - \alpha \Sigma \dot{z}.$$

This equation describes a damped harmonic motion. The solutions have the form $z(t) = A\exp(-\frac{t\alpha}{2}\sqrt{\Sigma})\sin(t\sqrt{(1 - \alpha^2/4)\Sigma} + \phi)$. Changing back to the variable $x$, we see that the iterates of the original method satisfy

$$x(t) = UA\exp(-\frac{t\alpha}{2}\sqrt{\Sigma})\sin(t\sqrt{(1 - \alpha^2/4)\Sigma} + \phi).$$

where $A$ and $\phi$ depend on the initialization.

From this analysis, we see that for small constant $\alpha$ the orbits of the lookahead method converge into the saddle point, and the error decays exponentially fast.

## A    PROOF OF THEOREM 1

Assume the optimal solution $(u^\star, v^\star)$ exists, then $\mathcal{L}'_u(u^\star, v) = \mathcal{L}'_v(u, v^\star) = 0$. In the following proofs, we use $g_u(u, v)$, $g_v(u, v)$ to represent the stochastic approximation of gradients, where $\mathbb{E}[g_u(u, v)] = \mathcal{L}'_u(u, v)$, $\mathbb{E}[g_v(u, v)] = \mathcal{L}'_v(u, v)$. We show the convergence of the proposed stochastic primal-dual gradients for the primal-dual gap $P(u^k, v^k) = \mathcal{L}(u^k, v^\star) - \mathcal{L}(u^\star, v^k)$. We prove the $O(1/\sqrt{k})$ convergence rate in Theorem 1 by using Lemma 1 and Lemma 2, which present the contraction of primal and dual updates, respectively.

**Lemma 1.** *Suppose $\mathcal{L}(u, v)$ is convex in $u$ and $\mathbb{E}[\|g_u(u, v)\|^2] \leq G_u^2$, we have*

$$\mathbb{E}[\mathcal{L}(u^k, v^k)] - \mathbb{E}[\mathcal{L}(u^\star, v^k)] \leq \frac{1}{2\alpha^k}\left(\mathbb{E}[\|u^k - u^\star\|^2] - \mathbb{E}[\|u^{k+1} - u^\star\|^2]\right) + \frac{\alpha_k}{2}G_u^2 \tag{16}$$

*Proof.* Use primal update in (3), we have

$$\|u^{k+1} - u^\star\|^2 = \|u^k - \alpha_k g_u(u^k, v^k) - u^\star\|^2 \tag{17}$$
$$= \|u^k - u^\star\|^2 - 2\alpha_k \langle g_u(u^k, v^k), u^k - u^\star \rangle + \alpha_k^2 \|g_u(u^k, v^k)\|^2. \tag{18}$$

Take expectation on both side of the equation, substitute with $\mathbb{E}[g_u(u, v)] = \mathcal{L}'_u(u, v)$ and apply $\mathbb{E}[\|g_u^2(u, v)\|] \leq G_u^2$ to get

$$\mathbb{E}[\|u^{k+1} - u^\star\|^2] \leq \mathbb{E}[\|u^k - u^\star\|^2] - 2\alpha_k \mathbb{E}[\langle \mathcal{L}'_u(u^k, v^k), u^k - u^\star \rangle] + \alpha_k^2 G_u^2. \tag{19}$$

Since $\mathcal{L}(u, v)$ is convex in $u$, we have

$$\langle \mathcal{L}'_u(u^k, v^k), u^k - u^\star \rangle \geq \mathcal{L}(u^k, v^k) - \mathcal{L}(u^\star, v^k). \tag{20}$$

(16) is proved by combining (19) and (20). $\qquad\square$

**Lemma 2.** *Suppose $\mathcal{L}(u, v)$ is concave in $v$ and has Lipschitz gradients, $\|\mathcal{L}'_v(u_1, v) - \mathcal{L}'_v(u_2, v)\| \leq L_v \|u_1 - u_2\|$; and bounded variance, $\mathbb{E}[\|g_u(u, v)\|^2] \leq G_u^2$, $\mathbb{E}[\|g_v(u, v)\|^2] \leq G_v^2$; and $\mathbb{E}[\|v^k - v^\star\|^2] \leq D_v^2$, we have*

$$\mathbb{E}[\mathcal{L}(u^k, v^\star)] - \mathbb{E}[\mathcal{L}(u^k, v^k)] \leq$$
$$\frac{1}{2\beta_k}\left(\mathbb{E}[\|v^k - v^\star\|^2] - \mathbb{E}[\|v^{k+1} - v^\star\|^2]\right) + \frac{\beta_k}{2}G_v^2 + \alpha_k L_v\left(G_u^2 + D_v^2\right). \tag{21}$$

*Proof.* From the dual update in (3), we have

$$\|v^{k+1} - v^\star\|^2 = \|v^k + \beta_k \, g_v(\bar{u}^{k+1}, v^k) - v^\star\|^2 \tag{22}$$

$$= \|v^k - v^\star\|^2 + 2\beta_k \, \langle g_v(\bar{u}^{k+1}, v^k), \, v^k - v^\star \rangle + \beta_k^2 \, \|g_v(\bar{u}^{k+1}, v^k)\|^2. \tag{23}$$

Take expectation on both sides of the equation, substitute $\mathbb{E}[g_v(u, v)] = \mathcal{L}'_v(u, v)$, and apply $\mathbb{E}[\|g_v^2(u, v)\|] \leq G_v^2$ to get

$$\mathbb{E}[\|v^{k+1} - v^\star\|^2] \leq \mathbb{E}[\|v^k - v^\star\|^2] + 2\beta_k \, \mathbb{E}[\langle \mathcal{L}'_v(\bar{u}^{k+1}, v^k), \, v^k - v^\star \rangle] + \beta_k^2 \, G_v^2. \tag{24}$$

Reorganize (24) to get

$$\mathbb{E}[\|v^{k+1} - v^\star\|^2] - \mathbb{E}[\|v^k - v^\star\|^2] - \beta_k^2 \, G_v^2 \leq 2\beta_k \, \mathbb{E}[\langle \mathcal{L}'_v(\bar{u}^{k+1}, v^k), \, v^k - v^\star \rangle]. \tag{25}$$

The right hand side of (25) can be represented as

$$\mathbb{E}[\langle \mathcal{L}'_v(\bar{u}^{k+1}, v^k), \, u^k - v^\star \rangle] \tag{26}$$

$$= \mathbb{E}[\langle \mathcal{L}'_v(\bar{u}^{k+1}, v^k) - \mathcal{L}'_v(u^k, v^k) + \mathcal{L}'_v(u^k, v^k), \, v^k - v^\star \rangle] \tag{27}$$

$$= \mathbb{E}[\langle \mathcal{L}'_v(\bar{u}^{k+1}, v^k) - \mathcal{L}'_v(u^k, v^k), \, v^k - v^\star \rangle] + \mathbb{E}[\langle \mathcal{L}'_v(u^k, v^k), \, v^k - v^\star \rangle], \tag{28}$$

where

$$\mathbb{E}[\langle \mathcal{L}'_v(\bar{u}^{k+1}, v^k) - \mathcal{L}'_v(u^k, v^k), \, v^k - v^\star \rangle] \tag{29}$$

$$\leq \mathbb{E}[\|\mathcal{L}'_v(\bar{u}^{k+1}, v^k) - \mathcal{L}'_v(u^k, v^k)\| \, \|v^k - v^\star\|] \tag{30}$$

$$\leq \mathbb{E}[L_v \, \|\bar{u}^{k+1} - u^k\| \, \|v^k - v^\star\|] \tag{31}$$

$$= \mathbb{E}[2L_y \, \|u^{k+1} - u^k\| \, \|v^k - v^\star\|] \tag{32}$$

$$= \mathbb{E}[2L_y \, \|\alpha_k g_u(u^k, v^k)\| \, \|v^k - v^\star\|] \tag{33}$$

$$\leq L_y \alpha_k \, \mathbb{E}[\|g_u(u^k, v^k)\|^2 + \|v^k - v^\star\|^2] \tag{34}$$

$$\leq L_y \alpha_k \, (G_u^2 + D_v^2). \tag{35}$$

Lipschitz smoothness is used for (31); the prediction step in (3) is used for (32); the primal update in (3) is used for (33); bounded assumptions are used for (35).

Since $\mathcal{L}(u, v)$ is concave in $v$, we have

$$\langle \mathcal{L}'_v(u^k, v^k), \, v^k - v^\star \rangle \leq \mathcal{L}(u^k, v^k) - \mathcal{L}(u^k, v^\star). \tag{36}$$

Combine equations (25, 28, 35 to get36)

$$\frac{1}{2\beta_k} \left( \mathbb{E}[\|v^{k+1} - v^\star\|^2] - \mathbb{E}[\|v^k - v^\star\|^2] \right) - \frac{\beta_k}{2} G_v^2$$
$$\leq L_v \alpha_k \, (G_u^2 + D_v^2) + \mathbb{E}[\mathcal{L}(u^k, v^k)] - \mathbb{E}[\mathcal{L}(u^k, v^\star)]. \tag{37}$$

Rearrange the order of (37) to achieve (21). $\qquad\square$

We now present the proof of Theorem 1.

*Proof.* Combining (16) and (21) in the Lemmas, the primal-dual gap $P(u^k, v^k) = \mathcal{L}(u^k, v^\star) - \mathcal{L}(u^\star, v^k)$ satisfies,

$$\mathbb{E}[P(u^k, v^k)] \leq \frac{1}{2\alpha_k} \left( \mathbb{E}[\|u^k - u^\star\|^2] - \mathbb{E}[\|u^{k+1} - u^\star\|^2] \right) + \frac{\alpha_k}{2} G_u^2$$
$$+ \frac{1}{2\beta_k} \left( \mathbb{E}[\|v^k - v^\star\|^2] - \mathbb{E}[\|v^{k+1} - v^\star\|^2] \right) + \frac{\beta_k}{2} G_v^2 + \alpha_k L_v \, (G_u^2 + D_v^2). \tag{38}$$

Accumulate (38) from $k = 1, \ldots, l$ to obtain

$$\sum_{k=1}^{l} \mathbb{E}[P(u^k, v^k)] \leq$$

$$\frac{1}{2\alpha_1} \mathbb{E}[\|u^1 - u^\star\|^2] + \sum_{k=2}^{l} \left( \frac{1}{2\alpha_k} - \frac{1}{2\alpha_{k-1}} \right) \mathbb{E}[\|u^k - u^\star\|^2] + \sum_{k=1}^{l} \alpha_k \left( \frac{G_u^2}{2} + L_v G_u^2 + L_v D_v^2 \right) \tag{39}$$

$$+ \frac{1}{2\beta_1} \mathbb{E}[\|v^1 - v^\star\|^2] + \sum_{k=2}^{l} \left( \frac{1}{2\beta_k} - \frac{1}{2\beta_{k-1}} \right) \mathbb{E}[\|v^k - v^\star\|^2] + \sum_{k=1}^{l} \beta_k \frac{G_v^2}{2}.$$

Assume $\mathbb{E}[\|u^k - u^\star\|^2] \leq D_u^2$, $\mathbb{E}[\|v^k - v^\star\|^2] \leq D_v^2$ are bounded, we have

$$\sum_{k=1}^{l} \mathbb{E}[P(u^k, v^k)] \leq \frac{1}{2\alpha_1} D_u^2 + \sum_{k=2}^{l} (\frac{1}{2\alpha_k} - \frac{1}{2\alpha_{k-1}}) D_u^2 + \sum_{k=1}^{l} \alpha_k (\frac{G_u^2}{2} + L_v G_u^2 + L_v D_v^2)$$
$$+ \frac{1}{2\beta_1} D_v^2 + \sum_{k=2}^{l} (\frac{1}{2\beta_k} - \frac{1}{2\beta_{k-1}}) D_v^2 + \sum_{k=1}^{l} \beta_k \frac{G_v^2}{2}. \tag{40}$$

Since $\alpha_k, \beta_k$ are decreasing and $\sum_{k=1}^{l} \alpha_k \leq C_\alpha \sqrt{l+1}$, $\sum_{k=1}^{l} \beta_k \leq C_\beta \sqrt{l+1}$, we have

$$\sum_{k=1}^{l} \mathbb{E}[P(u^k, v^k)] \leq \frac{\sqrt{l}}{2} \left( \frac{D_u^2}{C_\alpha} + \frac{D_v^2}{C_\beta} \right) + \sqrt{l+1} \left( \frac{C_\alpha G_u^2}{2} + C_\beta L_v G_u^2 + C_\alpha L_v D_v^2 + \frac{C_\beta G_v^2}{2} \right) \tag{41}$$

For $\hat{u}^l = \frac{1}{l} \sum_{k=1}^{l} u^k$, $\hat{v}^l = \frac{1}{l} \sum_{k=1}^{l} v^k$, because $\mathcal{L}(u, v)$ is convex-concave, we have

$$\mathbb{E}[P(\hat{u}^l, \hat{v}^l)] = \mathbb{E}[\mathcal{L}(\hat{u}^l, v^\star) - \mathcal{L}(u^\star, \hat{v}^l)] \tag{42}$$

$$\leq \mathbb{E}[\frac{1}{l} \sum_{k=1}^{l} (\mathcal{L}(u^k, v^\star) - \mathcal{L}(u^\star, v^k))] \tag{43}$$

$$= \frac{1}{l} \sum_{k=1}^{l} \mathbb{E}[\mathcal{L}(u^k, v^\star) - \mathcal{L}(u^\star, v^k)] \tag{44}$$

$$= \frac{1}{l} \sum_{k=1}^{l} \mathbb{E}[P(u^k, v^k)]. \tag{45}$$

Combine (41) and (45) to prove

$$\mathbb{E}[P(\hat{x}^l, \hat{y}^l)] \leq \frac{1}{2\sqrt{l}} \left( \frac{D_u^2}{C_\alpha} + \frac{D_v^2}{C_\beta} \right) + \frac{\sqrt{l+1}}{l} \left( \frac{C_\alpha G_u^2}{2} + C_\alpha L_v G_u^2 + C_\alpha L_v D_v^2 + \frac{C_\beta G_v^2}{2} \right). \tag{46}$$

$\square$

## B  MNIST TOY EXAMPLE

**Experimental details**: We consider a classic MNIST digits dataset (LeCun et al., 1998) consisting of 60,000 training images and 10,000 testing images each of size $28 \times 28$. For simplicity, let us consider a task (T1) of classifying into odd and even numbered images. Let's say, that $\sim 50\%$ of data instances were corrupted using salt and pepper noise of probability 0.2 and this distortion process was biased. Specifically, only 30% of even numbered images were distorted as against the 70% of odd-numbered images. We have observed that any feature representation network $\theta_f$ trained using the binary classification loss function for task T1 has noise bias also encoded within it. This was verified by training an independent noise classifier on the learned features. This lead us to design of simple adversarial network to "unlearn" the noise bias from the feature learning pipeline. We formulate this using the minimax objective in (5).

In our model, $\mathcal{L}_d$ is a softmax loss for the task (T2) of classifying whether the input sample is noisy or not. $\mathcal{L}_y$ is a softmax loss for task T1 and $\lambda = 1$. A LeNet network (LeCun et al., 1998) is considered for training on task T1 while a two-layer MLP is used for training on task T2. LeNet consist of two convolutional (conv) layers followed by two fully connected (FC) layers at the top. The parameters of conv layers form $\theta_f$ while that of FC and MLP layers forms $\theta_y$ and $\theta_d$ respectively. We train the network in three stages. Following the training on task T1, $\theta_f$ were fixed and MLP is trained independently on task T2. The default learning schedule of the LeNet implementation in Caffe (Jia et al., 2014) were followed for both the tasks. The total training iterations on each task were set to $10,000$. After pretraining, the whole network is jointly finetuned using the adversarial approach. (5) is alternatively minimized w.r.t. $\theta_\mathbf{f}, \theta_\mathbf{y}$ and maximized w.r.t. $\theta_\mathbf{d}$. The predictive steps were only used during the finetuning phase.

Our finding is summarized in Figure 3. In addition, Figure 7 provides head-to-head comparison of two popular solvers Adam and SGD using the predictive step. Not surprisingly, the Adam solver shows relatively better performance and convergence even with an additional predictive step. This also suggests that the default hyper-parameter for the Adam solver can be retained and utilized for training this networks without resorting to any further hyper-parameter tuning (as it is currently in practice).

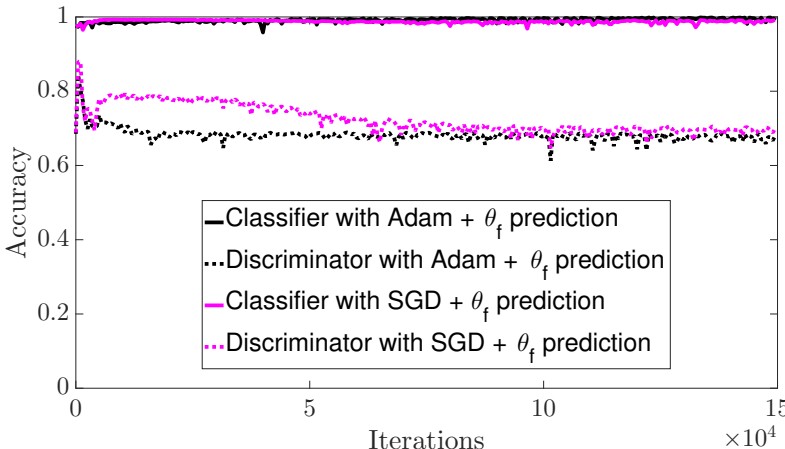

Figure 7: Comparison of the classification accuracy of parity classification and noise discrimination using the SGD and Adam solvers with and without prediction step.

## C   DOMAIN ADAPTATION

**Experimental details**: To evaluate a domain adaptation task, we consider the OFFICE dataset (Saenko et al., 2010). OFFICE is a small scale dataset consisting of images collected from three distinct domains: AMAZON, DSLR and WEBCAM. For such a small scale dataset, it is non-trivial to learn features from images of a single domain. For instance, consider the largest subset AMAZON, which contains only 2,817 labeled images spread across 31 different categories. However, one can leverage the power of domain adaptation to improve cross domain accuracy. We follow the protocol listed in Ganin & Lempitsky (2015) and the same network architecture is used. Caffe (Jia et al., 2014) is used for implementation. The training procedure from Ganin & Lempitsky (2015) is kept intact except for the additional prediction step. In Table 2 comparisons are drawn with respect to target domain accuracy on three pairs of source-target domain tasks. The test accuracy is reported at the end of 50,000 training iterations.

## D   FAIR CLASSIFIER

**Experimental details**: The "Adult" dataset from the UCI machine learning repository is used, which consists of census data from $\sim 45,000$ people. The task is to classify whether a person earns $\geq \$50k$/year. The person's gender is chosen to be the sensitive variable. We binarize all the category attributes, giving us a total of 102 input features per sample. We randomly split data into 35,000 samples for training, 5000 for validation and 5000 for testing. The result reported here is an average over five such random splits.

## E   GENERATIVE ADVERSARIAL NETWORKS

**Toy Dataset**: To illustrate the advantage of the prediction method, we experiment on a simple GAN architecture with fully connected layers using the toy dataset. The constructed toy example and its architecture is inspired by the one presented in Metz et al. (2017). The two dimensional data is sampled from the mixture of eight Gaussians with their means equally spaced around the unit circle

centered at $(0, 0)$. The standard deviation of each Gaussian is set at $0.01$. The two dimensional latent vector $\mathbf{z}$ is sampled from the multivariate Gaussian distribution. The generator and discriminator networks consist of two fully connected hidden layers, each with $128$ hidden units and tanh activations. The final layer of the generator has linear activation while that of discriminator has sigmoid activation. The solver optimizes both the discriminator and the generator network using the objective in (4). We use adam solver with its default parameters (i.e., learning rate = $0.001$, $\beta_1 = 0.9$, $\beta_2 = 0.999$) and with input batch size of $512$. The generated two dimensional samples are plotted in the figure (8). The straightforward utilization of the adam solver fails to construct all the modes of the underlying dataset while both unrolled GAN and our method are able to produce all the modes.

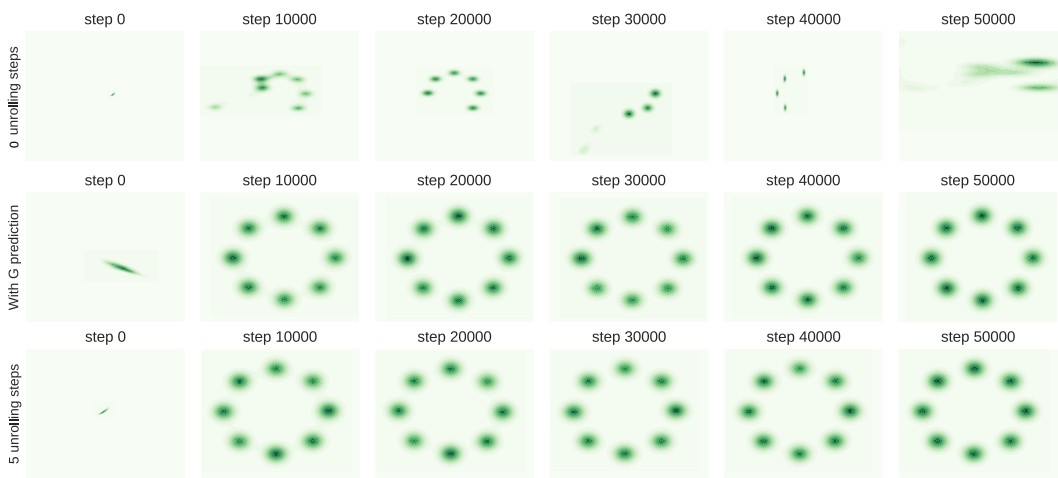

Figure 8: Comparison of GAN training algorithms on toy dataset. Results on, from top to bottom, GAN, GAN with **G** prediction, and unrolled GAN.

We further investigate the performance of GAN training algorithms on data sampled from a mixture of a large number of Gaussians. We use $100$ Gaussian modes which are equally spaced around a circle of radius $24$ centered at $(0, 0)$. We retain the same experimental settings as described above and train GAN with two different input batch sizes, a small $(64)$ and a large batch $(6144)$ setting. The Figure (9) plots the generated sample output of GAN trained (for fixed number of epochs) under the above setting using different training algorithms. Note that for small batch size input, the default as well as the unrolled training for GAN fails to construct actual modes of the underlying dataset. We hypothesize that this is perhaps due to the batch size, $64$, being smaller than the number of input modes $(100)$. When trained with small batch the GAN observe samples only from few input modes at every iteration. This causes instability leading to the failure of training algorithms. This scenario is pertinent to real datasets wherein the number of modes are relatively high compared to input batch size.

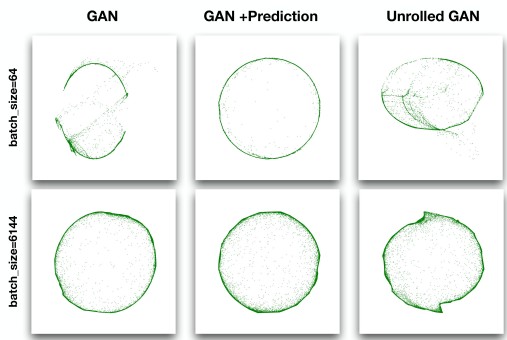

Figure 9: Comparison of GAN training algorithms on toy dataset of mixture of $100$ Gaussians. Results on, from top to bottom, batch size of $64$ and $6144$.

**DCGAN Architecture details**: For our experiments, we use publicly available code for DC-GAN (Radford et al., 2016) and their implementation for Cifar-10 dataset. The random noise vector is of $100$ dimensional and output of the generator network is a 64x64 image of 3 channels.

**Additional DCGAN Results**:

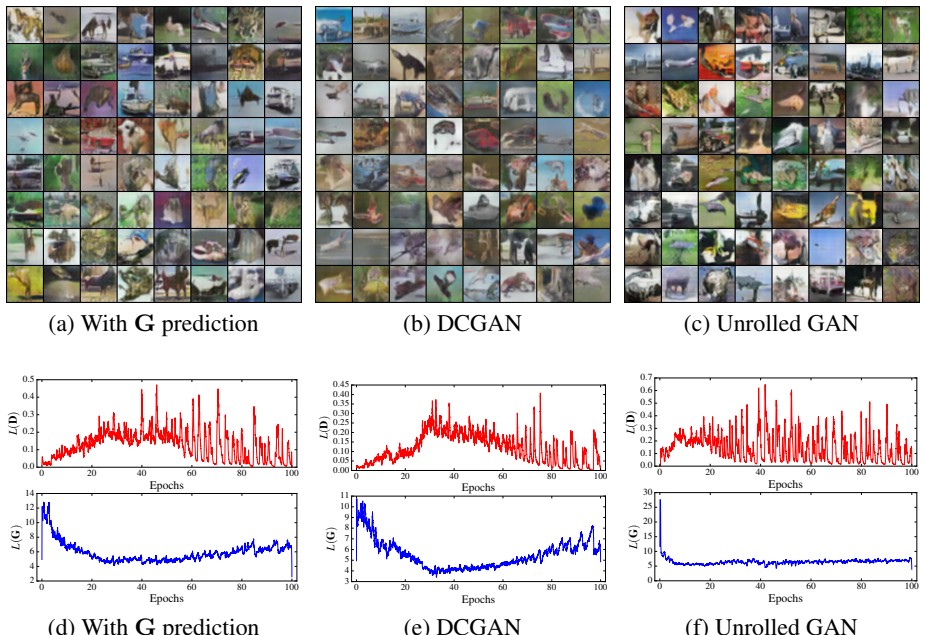

Figure 10: Comparison of GAN training algorithms for DCGAN architecture on Cifar-10 image datasets. Using higher momentum, $lr = 0.0002, \beta_1 = 0.9$.

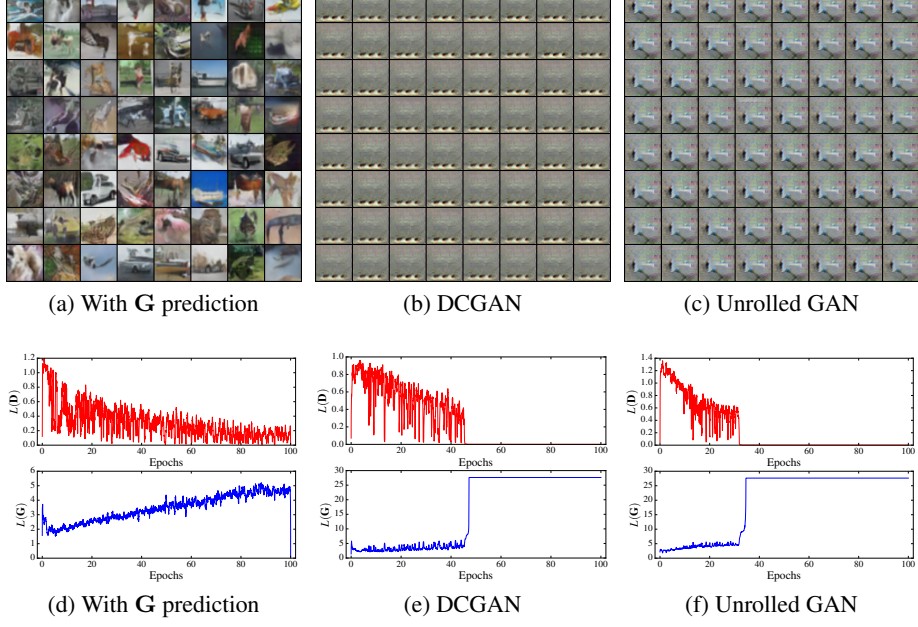

Figure 11: Comparison of GAN training algorithms for DCGAN architecture on Cifar-10 image datasets. $lr = 0.0004, \beta_1 = 0.5$.

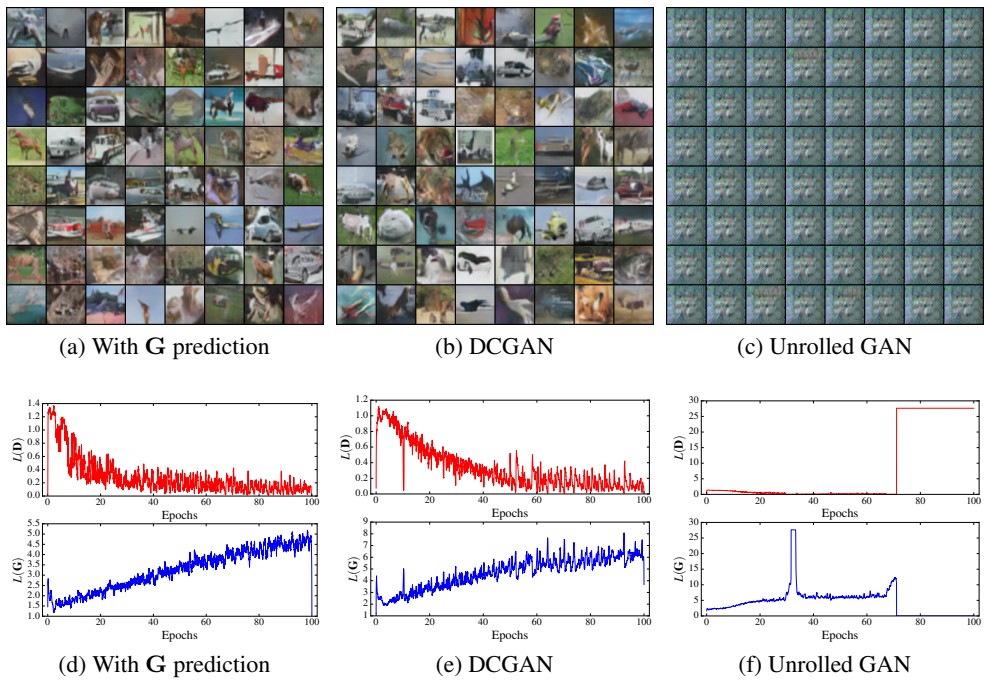

Figure 12: Comparison of GAN training algorithms for DCGAN architecture on Cifar-10 image datasets. $lr = 0.0006, \beta_1 = 0.5$.

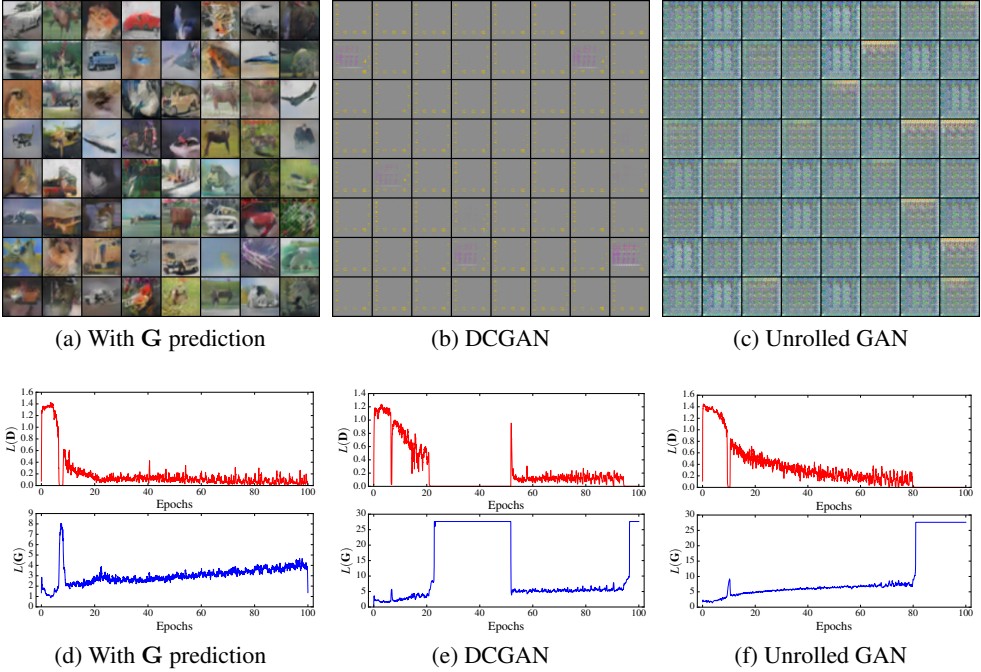

Figure 13: Comparison of GAN training algorithms for DCGAN architecture on Cifar-10 image datasets. $lr = 0.0008, \beta_1 = 0.5$.

**Experiments on Imagenet**: In this section we demonstrate the advantage of prediction methods for generating higher resolution images of size 128 x 128. For this purpose, the state-of-the-art AC-GAN (Odena et al., 2017) architecture is considered and conditionally learned using images of all 1000 classes from Imagenet dataset. We have used the publicly available code for AC-GAN and all the parameter were set to it default as in Odena et al. (2017). The figure 14 plots the inception score measured at every training epoch of AC-GAN model with and without prediction. The score is averaged over five independent runs. From the figure, it is clear that even at higher resolution with large number of classes the prediction method is stable and aids in speeding up the training.

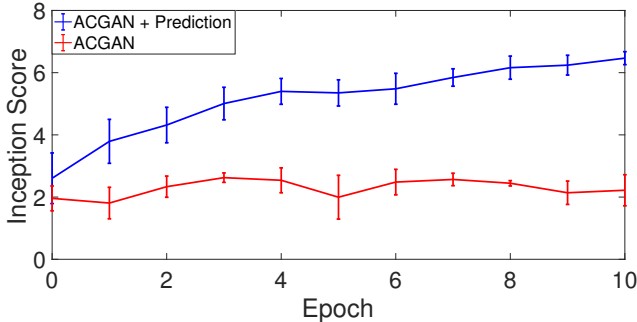

Figure 14: Comparison of Inception scores on high resolution Imagenet datasets measured at each training epoch of ACGAN model with and without prediction.

