# OpenReview forum: "Stabilizing Adversarial Nets with Prediction Methods"
_ICLR.cc/2018/Conference — Accept (Poster)_

### Official Review · AnonReviewer2 · 2017-11-26
**A simple modification to alternating stochastic gradient for GAN training, which stabilizes training, essentially for free. Clever and useful idea, solid and insightful analysis, good presentation.**

**Rating:** 9
**Confidence:** 4

**Review:**

This paper proposes a simple modification to the standard alternating stochastic gradient method for GAN training, which stabilizes training, by adding a prediction step.

This is a clever and useful idea, and the paper is very well written. The proposed method is very clearly motivated, both intuitively and mathematically, and the authors also provide theoretical guarantees on its convergence behavior. I particularly liked the analogy with the damped harmonic oscillator.

The experiments are well designed and provide clear evidence in favor of the usefulness of the proposed technique. I believe that the method proposed in this paper will have a significant impact in the area of GAN training.

I have only one minor question: in the prediction step, why not use a step size, say
$\bar{u}_k+1 = u_{k+1} + \gamma_k (u_{k+1} − u_k)$, such that the "amount of predition" may be adjusted?

---

> ### Author Response · Authors · 2017-12-28
> **Regarding step size gamma**
>
> Thanks for the thoughtful comments!  To answer your question:  it is indeed possible to generalize this method by adding an extra stepsize parameter for the prediction step, and this is something that we have experimented with extensively.  It can be shown that your proposed “gamma” parameter method is stable (under convexity assumptions) whenever gamma is between 0 and 2.  However, we have not been able to find any worthwhile advantages to choosing any gamma different from 1.  Choosing a smaller gamma weakens the stability benefits of prediction, and choosing a larger gamma seems to slow down convergence a bit.  The latter effect can be compensated for by choosing a larger learning rate, but even in this case the method doesn’t run noticeably faster than with gamma=1.  For this reason, including this “gamma” seemed like unnecessarily added complexity, so we removed it and went with a cleaner presentation.

---

### Official Review · AnonReviewer1 · 2017-11-26
**Good work**

**Rating:** 7
**Confidence:** 4

**Review:**

This work proposes a framework for stabilizing adversarial nets using a prediction step. The prediction step is motivated by primal-dual algorithms in convex optimization where the term having both variables is bi-linear.

The authors prove a convergence result when the function is convex in one variable and concave in the other. This problem is more general than the previous one in convex optimization.  Then this prediction step is applied in many recent applications in training adversarial nets and compared with state-of-the-art solvers. The better performance of this simple step is shown in most of the numerical experiments.

Though this work applies one step from the convex optimization to solve a more complicated problem and obtain improved performance,  there is more work to be done. Whether there is a better generalization of this prediction step? There are also other variants of primal-dual algorithms in convex optimization; can other modification including the accelerated variants be applied?

---

> ### Author Response · Authors · 2017-12-28
> **Regarding accelerated methods**
>
> We thank the reviewer for the thoughtful comments and suggestions for future work.   We think the idea of pursuing accelerated methods is particularly interesting.  We have actually already done some experiments with Nesterov-type acceleration (as described for saddle-point problems by Chambolle and Pock), however it seems that the benefits of acceleration vanish when we move from deterministic to stochastic updates.  We’ve made similar observations for standard convex (non-saddle) problems.  That being said, we’re still interested in this direction, and are keeping our eyes peeled for possible ways forward.

---

### Official Review · AnonReviewer4 · 2017-12-06
**Some issues with both experiments and theoretical claims.**

**Rating:** 4
**Confidence:** 4

**Review:**

NOTE:
I'm very willing to change my recommendation if I turn out to be wrong
about the issues I'm addressing and if certain parts of the experiments are fixed.

Having said that, I do (think I) have some serious issues:
both with the experimental evaluation and with the theoretical results.
I'm pretty sure about the experimental evaluation and less sure about the theoretical results.


THEORETICAL CLAIMS:

These are the complaints I'm not as sure about:

Theorem 1 assumes that L is convex/concave.
This is not generally true for GANs.
That's fine and it doesn't necessarily make the statement useless, but:

If we are willing to assume that L is convex/concave,
then there already exist other algorithms that will provably converge
to a saddle point (I think). [1] contains an explanation of this.
Given that there are other algorithms with the same theoretical guarantees,
and that those algorithms don't magically make GANs work better,
I am much less convinced about the value of your theorem.

In [0] they show that GANs trained with simultaneous gradient descent are locally asymptotically stable,
even when L is not convex/concave.
This seems like it makes your result a lot less interesting, though perhaps I'm wrong to think this?

Finally, I'm not totally sure you can show that simultaneous gradient descent won't converge
as well under the assumptions you made.
If you actually can't show that, then the therom *is* useless,
but it's also the thing I've said that I'm the least sure about.


EXPERIMENTAL EVALUATION:

Regarding the claims of being able to train with a higher learning rate:
I would consider this a useful contribution if it were shown that (by some measure of GAN 'goodness')
a high goodness was achieved faster because a higher learning rate was used.
Your experiments don't support this claim presently, because you evaluate all the models at the same step.
In fact, it seems like both evaluated Stacked GAN models get worse performance with the higher learning rate.
This calls into question the usefulness of training with a higher learning rate.
The performance is not a huge amount worse though (based on my understanding of Inception Scores),
so if it turns out that you could get that performance
in 1/10th the time then that wouldn't be so bad.

Regarding the experiment with Stacked GANs, the scores you report are lower than what they report [2].
Their reported mean score for joint training is 8.59.
Are the baseline scores you report from an independent reproduction?
Also, the model they have trained uses label information.
Does your model use label information?
Given that your reported improvements are small, it would be nice to know what the proposed mechanism is by
which the score is improved.
With a score of 7.9 and a standard deviation of 0.08, presumably none of the baseline model runs
had 'stability issues', so it doesn't seem like 'more stable training' can be the answer.

Finally, papers making claims about fixing GAN stability should support those claims by solving problems
with GANs that people previously had a hard time solving (due to instability).
I don't believe this is true of CIFAR10 (especially if you're using the class information).
See [3] for an example of a paper that does this by generating 128x128 Imagenet samples with a single generator.

I didn't pay as much attention to the non-GAN experiments because
a) I don't have as much context for evaluating them, because they are a bit non-standard.
b) I had a lot of issues with the GAN experiments already and I don't think the paper should be accepted unless those are addressed.


[0] https://arxiv.org/abs/1706.04156 (Gradient Descent GAN Optimization is Locally Stable)

[1] https://arxiv.org/pdf/1705.07215.pdf (On Convergence and Stability of GANs)

[2] https://arxiv.org/abs/1612.04357 (Stacked GAN)

[3] https://openreview.net/forum?id=B1QRgziT (Spectral Regularization for GANs)

EDIT:
As discussed below, I have slightly raised my score.
I would raise it more if more of my suggestions were implemented (although I'm aware that the authors don't have much (any?) time for this - and that I am partially to blame for that, since I didn't respond that quickly).
I have also slightly raised my confidence.
This is because now I've had more time to think about the paper, and because the authors didn't really address a lot of my criticisms (which to me seems like evidence that some of my criticisms were correct).

---

> ### Author Response · Authors · 2017-12-28
> **Response for theoretical comments**
>
> We agree with the reviewer that theory in this area (and in deep learning in general) often requires assumptions that don’t hold for neural networks.  Nonetheless, we think it is worth taking time to explore conditions under which algorithms are guaranteed to work, because this provides a theoretical proof-of-concept, and thinking through theoretical properties of a new algorithm makes it more than just another hack.   The purpose of our result is to do just that for our proposed algorithm.  We don’t disagree that analysis exists for other algorithms, but we don’t think the existence of other algorithms gets us “off the hook” from thinking about the theoretical implications of our approach.
>
> That being said, we think the reviewer is overestimating the state of the art in theory for GANs.  There is currently no theoretical result that does not make strong assumptions, and many results (including those referenced by the reviewer) are quite different from (and in many ways weaker than) our own.  The result in [1] shares certain assumptions with our own (convex-concave assumptions, bounded problem domain, and an ergodic measure of convergence).  However, the result in [1] does not prove convergence in the usual sense, but rather that the error will decay to within an o(1) constant.  In contrast, our result shows that the error decays to zero.  The result in [1] also requires simultaneous gradient descent, which is not commonly used in practice (because it requires more RAM to store [extremely large] iterates and it uses a stale iterate when updating the generator and discriminator one-at-a-time).  In contrast, our result concerns the commonly used alternating direction approach.
>    The result in [0] shows stability using a range of assumptions that are different from (but not necessarily stronger or weaker than) our own.  They require the discriminator to be a linear classifier, and make a strict concavity assumption on the loss function.  They also require an assumption (called Property I) that is analogous to the “strict saddle” assumption in the saddle-point literature (see, e.g. Lee 2016, “Gradient Descent Converges to Minimizers”), which is known not to hold for general neural nets.  Also, note that the result in [0] is only a local stability result (it only holds once the iterates get close to a saddle satisfying the assumptions), whereas our result is a global convergence result that holds for any initialization.
> 	Finally, we emphasize that both [0] and [1] are great works that make numerous important contributions to this field and address a host of issues beyond just convergence proofs.  Our purpose here is not to make any claims that our result is “better” than theirs, but rather to state what differentiates our result from the literature, and why we felt it was worth putting it in the paper.

---

> > ### Comment · AnonReviewer4 · 2018-01-13
> > **Response to both author comments**
> >
> > REGARDING THEORY RESPONSE:
> > I'm not specifically criticizing the lack of realism in the assumptions - I agree that such a criticism would be unreasonable.
> > Rather, I'm saying that other algorithms have similar theoretical guarantees (proved using similar assumptions) to your algorithm, yet those guarantees don't seem to correspond in general to serious improvements in empirical performance.
> > Thus, I estimate that the value of your particular guarantee is low.
> > If instead it were true that all GAN training procedures proven to have property P seem to do really well in practice, and you proved that your (admittedly simple and easy to implement) algorithm had property P, I would feel differently.
> >
> > You also didn't address my complaint that the proof of Theorem 1 seems like it would apply to simultaneous gradient descent (another commenter has now also made this claim).
> > This seems like an easy complaint to address - either I'm right about this or I'm wrong.
> > If I'm right, I don't see what value Theorem 1 adds (except as a sanity check), if I'm wrong, I'm happy to increase the score.
> >
> > Finally, I'm confused by your statements about simultaneous gradient descent.
> > Why should I expect it to use more significantly more RAM?
> > Perhaps we are talking about different things when we say Simultaneous Gradient Descent (maybe there is a usage in the saddle-point optimization literature I'm not familiar with)?
> >
> > REGARDING EXPERIMENTAL RESPONSE:
> >
> > > prediction makes really difficult problems really easy
> > Right - I claim that you haven't showed this.
> > All the problems you solved (admittedly with the exception of the 100 Mixture Components - but I can think of other methods that could solve this problem even more easily :)) have already been 'solved' in my estimation.
> > Your measure of 'really difficult' is  behind the GAN literature in an empirical sense.
> > I would also claim that the experiments where you've modified the hyperparameters in a variety of ways and shown that things are generally better behaved have already been done, e.g. in the Improved WGAN paper.
> > That doesn't necessarily mean that doing them again is useless, but certainly it decreases their value.
> >
> > > below we also report the performance score measured at the fewer number of epochs for higher learning rates.
> > I think this is much better empirical support for your method than anything else in the paper.
> > I will raise my score slightly for this reason, and I would raise it even more for a version that de-emphasized the claims to have SOTA inception score and more heavily explored this benefit (I'm aware you might not have time for that).
> >
> > I don't really know what to make of the Imagenet experiment.
> > For one thing, you have error bars, but surely you've conducted only one instance of the experiment (presumably all of your baseline runs didn't dip and then recover exactly at epoch 5)?
> >
> > MISC:
> >
> > Here are ways that I think this paper could be improved (which you are of course free to disregard):
> >
> > 1. Move Thm 1 to an appendix - I don't think it really does anything.
> > 2. Get rid of the bit about the oscillator - I don't think it's wrong per se, but the relevance is questionable.
> > The pathologies you're claiming to get rid of correspond more to divergence than to well-behaved cycling about a fixed point.
> > 3. Emphasize the speed-up you can get from using higher learning rates. This is a good result!
> > 4. Do proper imagenet experiments. I know they're a pain, but the state of the art has moved there at this point.
> > 5. Give a more complete story about why your method should prevent certain pathologies, and maybe study more deeply the nature of those pathologies? (I don't have a great suggestion about how to do this because I don't know what the story is!)

---

> > > ### Author Response · Authors · 2018-01-17
> > > **Response to reviewer concerns**
> > >
> > > > Rather, I'm saying that other algorithms have similar theoretical guarantees
> > > Could the reviewer be more specific on which algorithms have similar theoretical guarantees with ours? We believe we have clearly distinguish our analysis from the references mentioned before. We would like to emphasize theoretical results (especially upper bound) only provides worst-case guarantee. The empirical performance may vary for different algorithms under same guarantees. Again, we agree GANs may not satisfy our assumptions that have been widely used in GAN optimization, but the analysis is not unnecessary.
> > >
> > > > the proof of Theorem 1 seems like it would apply to simultaneous gradient descent
> > > The main purpose of theorem 1 is to show the prediction method converges in a convex-concave setting. It is correct that similar analysis can be applied to general alternating gradients and simultaneous gradients without the prediction step. However, we are unaware of previous convergence analysis for alternative gradients with prediction step except for the (non-stochastic) bilinear problems discussed in related work.
> > >
> > > > Why should I expect it to use more significantly more RAM?
> > > In most current implementation frameworks, we need to store weights of generator, weights of discriminator, gradients of generator, and gradients of discriminator, and  all the four variables need to be stored in RAM (GPU memory) for simultaneous gradient descent. However, only one of the gradients (either generator or discriminator) needs to be stored in GPU anytime for alternate updates. It can make a big difference for training large networks on GPU with limited memory.
> > >
> > > >> prediction makes really difficult problems really easy
> > > The reviewer is simply nitpicking the part of our response. In our earlier response we have clearly quoted,
> > >
> > > “The purpose of the experiments is not to show that we can train things that are *impossible* to train via other methods (indeed, almost anything is possible if you tune the hyperparameters and network architecture enough), but rather that prediction makes really difficult problems really easy.”
> > >
> > > This do suggest that the models considered in our work can be solved by various methods. However, each of these models requires different tricks to actually make them work (For more details please refer section 3.2). In our work, three different GAN models were considered. For each of these models, single method i.e., prediction has been shown to work equally well for the default setting and remains stable for wide range of hyper-parameters. Moreover, it is not clear whether the tricks mentioned in Improved WGAN paper works well when applied to other GAN models or loss functions.
> > >
> > > > Your measure of 'really difficult' is  behind the GAN literature in an empirical sense.
> > > Could the reviewer be more specific ? Pointer to any references ?
> > >
> > > Regarding Imagenet:
> > > The reported variance was the outcome of the inception score code. In our latest revision, the updated  figure is now an average over five different instances.
> > >
> > > We also thank reviewer for suggestions on improving our paper presentation.

---

> ### Author Response · Authors · 2017-12-28
> **Experimental comments**
>
> The purpose of the experiments is not to show that we can train things that are *impossible* to train via other methods (indeed, almost anything is possible if you tune the hyperparameters and network architecture enough), but rather that prediction makes really difficult problems really easy.  Compared to simple alternating gradient methods, prediction methods are more stable than other methods, work with a much wider range of hyperparameters than classical schemes, and don’t suffer from the collapse phenomenon that make it difficult to use other methods.
>
>  Below, we address reviewer’s comments that seem to pertain to specific dataset and architecture:
>
> Regarding DCGAN experiments (Without using label information):
> Figure 4 uses the finely tuned learning rate and momentum parameters that come with the pre-packaged DCGAN code distribution. This figure shows that DCGAN collapses frequently; even with these fine tuned parameters it still requires a carefully chosen stopping time/epoch to avoid collapse.  With prediction it does not collapse at all.  The purpose of increasing the learning rate is not to show that “better” results could be had, but rather to show that prediction methods don’t require finely tuned parameters.  If you have a look at the additional experiments in the appendix (page 18), we train DCGAN with a litany of different learning rate and momentum parameters.  The prediction method succeeds without any collapse events in all of these cases, while non-prediction is unstable as soon as we move away from the carefully tuned parameter choices.
>
> Regarding Stacked GAN (With using label information):
>    We reproduced this experiment using the Stacked Gan author’s publicly available code, but were not able to get the same inception scores for Stacked GAN as the original authors.  Note the release code did not come with code for computing inception scores, and we used a well-known Tensor Flow implementation that may differ from what the original author’s used.
>    We ran all the scenarios for a fixed number of epochs (200 epochs, which is default in the Stacked GAN’s released code) to ensure a fair comparison. Indeed, prediction method was able to achieve the best inception score of 8.83 at lesser epoch than 200. Having said that, as per the suggestion, below we also report the performance score measured at the fewer number of epochs for higher learning rates. The quantitative comparison based on the inception score for learning rates of 0.0005 (200/5 = 40 epochs) and 0.001 (200/10 = 20 epochs) are as follows-
>
> Learning Rate			              0.0005 (epochs=40)		         0.001 (epochs=20)
> Stacked GAN (joint)                                 5.80 +/- 0.15                                    1.42 +/- 0.01
> Stacked GAN (joint) + Prediction           8.10 +/- 0.10                                    7.79 +/- 0.07
>
> Regarding the absence of problems that are “hard” without prediction:  In Figure 8 of the appendix, we solve a toy problem that is famously hard:  trying to recover all of the modes in a Gaussian mixture model.  The prediction method does this easily, while the method without prediction fails to capture all the modes.  We also “turn the dial up” on this problem by using 100 Gaussian components in Figure 9, and the non-prediction method produces highly irregular results unless a batch size of over 6000 (which is very much larger than the number of components) is used.  In contrast, the prediction method represents the distribution well for a wide range of batch sizes and learning rates.

---

> > ### Author Response · Authors · 2018-01-04
> > **Results on Imagenet**
> >
> > As per the suggestion, we experimented with Imagenet dataset using AC-GAN [1] model with and without prediction. Please find the result in the supplementary material. Note that unlike the model used in spectral regularization [2] article, AC-GAN model do not use conditional BN, resnet blocks, hinge loss etc. Thus compared to [2], the reported inception score is low. We stick to AC-GAN as it is the only publicly available model which works best on Imagenet dataset.
> >
> > [1] https://arxiv.org/abs/1610.09585 (AC-GAN)
> > [2] https://openreview.net/forum?id=B1QRgziT (Spectral Regularization for GANs)

---

> ### Author Response · Authors · 2018-01-21
> **Any remaining Concerns ?**
>
> Dear Reviewer,
>
> We have tried addressing all your concerns in our latest response, please let us know if you still have any remaining concerns ?

---

### Public Comment · (anonymous) · 2017-12-06
**Question regarding Theorem 1**

It seems to me that the proof of Theorem 1 would go through even without the prediction step, i.e. it would be equally valid for, say, simultaneous gradient descent (instead of Lemma 2, one only needs to use a similar argument as in Lemma 1). In that sense, the theorem provides no support for the proposed method.

---

> ### Public Comment · ~Leon_Boellmann1 · 2017-12-07
> **faster convergence**
>
> I just pass by and see this question. This paper is actually very close to what I have been doing (not published yet), so I would like to share some of my understandings.
>
> The convergence rate for simultaneous gradient descent is generally slow and with the prediction step the order of the convergence rate could be increased. Details can be found in the paper in the reference list: Chen et al, "Optimal primal-dual methods for a class of saddle point problems".
>
> Actually I think this is a very good paper (probably because I am doing very similar work). Although one of the reviewers gave a very low score, these questions are very reasonable and are what I have in mind too. I believe the authors should be aware of them too. I hope the authors can justify the contributions of the paper.
>
> For the convex-concave assumption, this paper is not the only one that assumes that. The f-GAN paper also derived a theorem based on this assumption. The recent paper "Training GANs with Optimism", which is submitted to this ICLR, also derives the main algorithm based on this assumption. They are still very excellent papers.

---

### Public Comment · (anonymous) · 2018-01-03
**A relevant reference**

Hi, just wanted to point out a very relevant paper from a different community.

https://link.springer.com/article/10.1134%2FS0965542511120050

I guess it deserves to be mentioned in your paper.

---

### Author Response · Authors · 2018-01-04
**Results on Imagenet**

As per the suggestion from Reviewer4, we experimented with Imagenet dataset using AC-GAN [1] model with and without prediction. Please find the result in the supplementary material. Note that unlike the model used in spectral regularization [2] article, AC-GAN model do not use conditional BN, resnet blocks, hinge loss etc. Thus compared to [2], the reported inception score is low. We stick to AC-GAN as it is the only publicly available model which works best on Imagenet dataset.

[1] https://arxiv.org/abs/1610.09585 (AC-GAN)
[2] https://openreview.net/forum?id=B1QRgziT (Spectral Regularization for GANs)

---

### Public Comment · (anonymous) · 2018-01-12
**Works for my non-standard Adversarial network architecture !!**

I would like to just add that I am working on a lighting related project which uses Adversarial network architecture. Because my architecture is bit non-standard GAN architecture, I was finding very difficult to train my network. But after implementing this simple idea (it look hardly 2 hours for me to code), at-least I am able to train my network and get some reasonable results (although far from what I wanted).

---

> ### Author Response · Authors · 2018-01-12
> **Thanks**
>
> Thanks for trying it out, please let me know if you need any help implementing it ?

---

### Author Response · Authors · 2018-01-12
**Any questions or concerns ?**

Dear ACs and Reviewers,

Do you have any questions?
Are there any remaining concerns?

Best regards,
The Authors

---

### Decision · Program_Chairs · 2018-01-29
**ICLR 2018 Conference Acceptance Decision**

**Decision:**

Accept (Poster)

**Comment:**

This paper provides a simple technique for stabilizing GAN training, and works over a variety of GAN models.

One of the reviewers expressed concerns with the value of the theory. I think that it would be worth emphasizing that similar arguments could be made for alternating gradient descent, and simultaneous gradient descent. In this case, if possible, it would be good to highlight how the convergence of the prediction method approach differs from the alternating descent approach. Otherwise, highlight that this theory simply shows that the prediction method is not a completely crazy idea (in that it doesn't break existing theory).

Practically, I think the experiments are sufficiently interesting to show that this approach has promise. I don't see the updated results for Stacked GAN for a fixed set of epochs (20 and 40 at different learning rates). Perhaps put this below Table 1.